# Genomic analyses provide insights into spinach domestication and the genetic basis of agronomic traits

Xiaofeng Cai [1,2,10], Xuepeng Sun[3,4,10], Chenxi Xu[1,10], Honghe Sun [3,5], Xiaoli Wang[1], Chenhui Ge[1], Zhonghua Zhang[6], Quanxi Wang [7✉], Zhangjun Fei [3,8✉], Chen Jiao [3,9,10✉] & Quanhua Wang [1✉]

Spinach is a nutritious leafy vegetable belonging to the family Chenopodiaceae. Here we report a high-quality chromosome-scale reference genome assembly of spinach and genome resequencing of 305 cultivated and wild spinach accessions. Reconstruction of ancestral Chenopodiaceae karyotype indicates substantial genome rearrangements in spinach after its divergence from ancestral Chenopodiaceae, coinciding with high repeat content in the spinach genome. Population genomic analyses provide insights into spinach genetic diversity and population differentiation. Genome-wide association studies of 20 agronomical traits identify numerous significantly associated regions and candidate genes for these traits. Domestication sweeps in the spinach genome are identified, some of which are associated with important traits (e.g., leaf phenotype, bolting and flowering), demonstrating the role of artificial selection in shaping spinach phenotypic evolution. This study provides not only insights into the spinach evolution and domestication but also valuable resources for facilitating spinach breeding.

[1] Shanghai Engineering Research Center of Plant Germplasm Resources, College of Life Sciences, Shanghai Normal University, 200234 Shanghai, China. [2] Qinghai Key Laboratory of Vegetable Genetics and Physiology, Qinghai University, 810016 Xining, China. [3] Boyce Thompson Institute, Cornell University, Ithaca, NY 14853, USA. [4] College of Agriculture and Food Science, Zhejiang A&F University, 311300 Hangzhou, China. [5] Plant Biology Section, School of Integrative Plant Science, Cornell University, Ithaca, NY 14853, USA. [6] Engineering Laboratory of Genetic Improvement of Horticultural Crops of Shandong Province, College of Horticulture, Qingdao Agricultural University, 266109 Qingdao, China. [7] College of Life Science and Technology, Harbin Normal University, 150025 Harbin, China. [8] USDA-ARS, Robert W. Holley Center for Agriculture and Health, Ithaca, NY 14853, 18, USA. [9] Key Lab of Molecular Biology of Crop Pathogens and Insects, Institute of Biotechnology, Zhejiang University, 310058 Hangzhou, China. [10]These authors contributed equally: Xiaofeng Cai, Xuepeng Sun, Chenxi Xu, Chen Jiao. ✉email: wangqx@shnu.edu.cn; zf25@cornell.edu; biochenjiao@zju.edu.cn; wangquanhua@shnu.edu.cn

Spinach (*Spinacia oleracea* L., $2n = 2 \times = 12$) is a highly nutritious leafy vegetable rich in vitamins and mineral elements, with a global production of 30.1 million tonnes in 2019 (FAOSTAT; http://faostat.fao.org) and the harvested acreage increasing over years[1]. Spinach belongs to the Chenopodiaceae family in the order Caryophyllales. A number of quantitative trait loci (QTLs), markers and genes associated with agronomic traits such as leaf morphology[2,3], bolting[4], flowering, and nutritional quality[5,6] have been identified for spinach during the past decade; however, there is still an urgent need for spinach improvement, for example, to reduce the concentration of oxalate that can cause health issues when excessively consumed[7], and to enhance the resistance to major diseases.

The spinach germplasm comprising cultivars and wild accessions display a wide array of morphological diversity, including that of important traits such as the contents of carotenoid, folate, oxalate, and nitrate[8], which provides a great potential for spinach improvement. Similar to many other crops, the production of spinach has been challenged by biotic stresses from diseases, pests and weed infestations, and abiotic stresses such as salinity, drought, and heat. In contrast to the abiotic stress tolerance and quality traits, which have received less attention in spinach research and breeding, resistance against diseases especially downy mildew has been the main target in spinach breeding. Introgression of nucleotide-binding site leucine-rich repeat (NBS-LRR) resistance genes from wild relatives is the major strategy to develop downy mildew resistant cultivars, whereas the use of loss-of-function alleles of susceptibility genes may provide a durable strategy to develop resistant cultivars[8].

Understanding the genetic variation of the germplasm is critical for facilitating spinach breeding. The genus *Spinacia* contains two wild species, *S. turkestanica* Ilj. and *S. tetrandra* Stev., with the former being considered as the direct progenitor of cultivated spinach[9]. Previously, we generated a draft genome of cultivated spinach and sequenced the transcriptomes of 120 cultivated and wild spinach accessions[9]. However, due to the technical limitations at that time, the genome assembly is largely fragmented and the identified genetic variations are restricted mainly to the transcribed regions. Recently, two additional draft genomes of spinach assembled using PacBio long reads and Illumina short reads have been released[10,11] (Spov3 and SOL_r1.1). However, a chromosome-level reference-grade genome for spinach is still urgently needed to facilitate comparative genomic, genetic mapping and gene cloning studies. In the present study, we report a much-improved chromosome-scale spinach reference genome assembled using PacBio long reads and chromatin interaction maps generated using the high-throughput chromosome conformation capture (Hi-C) technology. The assembly is highly accurate, complete, and continuous, with ~98.3% anchored and ordered on the six spinach chromosomes. We reconstruct the ancestral karyotype of Chenopodiaceae to elucidate its evolution history. We then construct a genome variation map by genome resequencing of 305 cultivated and wild spinach accessions. We perform genome-wide association studies (GWAS) to dissect the genetic architecture of important spinach traits, and conduct population genomic analyses to elucidate the history of spinach breeding.

## Results

### Chromosome-scale reference genome assembly of spinach.
An inbred line Monoe-Viroflay, developed by recurrent selfing of a single monoecious plant for more than 10 generations, was used for reference genome sequencing. The line is a derivative of an XX female individual. We generated 110 Gb (118×) PacBio CLR long reads and 102 Gb (109×) Illumina short reads (Supplementary Table 1). Analysis of *k*-mer distribution using Illumina reads showed only a homozygous peak in Monoe-Viroflay, whereas the heterozygous peak which was observed in the previously sequenced sibling inbred line Sp75 was absent (Fig. 1a), confirming the high homozygosity of the Monoe-Viroflay inbred line. The PacBio long reads were self-corrected and then de novo assembled into contigs, which were further polished using Illumina paired-end reads. The assembled contigs were clustered into six pseudomolecules using both short- and long-range chromatin interaction maps derived from Dovetail Chicago and Hi-C data, respectively (Fig. 1b and Supplementary Data 1). Five contigs were identified to contain misassemblies and thus broken up based on the evidence of both Hi-C contact signals and the published genetic maps[9] (Supplementary Fig. 1). The final assembly comprised 307 contigs with a total size of ~894.3 Mb and an N50 contig size of ~23.8 Mb. The contig N50 size of the Monoe-Viroflay assembly was three orders of magnitude larger than that of Sp75 (16.6 kb)[9] and also much larger than that of Spov3 (1.8 Mb)[10] and SOL_r1.1 (11.3 Mb)[11] (Supplementary Data 1). Approximately 98.3% of assembled contigs were anchored and ordered on the chromosomes (Fig. 1c and Supplementary Fig. 2), which doubled the size of the anchored sequences in the Sp75 genome assembly (46.5%) and were also much higher than that in the Spov3 (81.6%) and SOL_r1.1 (73.6%) assemblies (Supplementary Data 1 and Supplementary Fig. 3). Assessment of the Monoe-Viroflay assembly using Merqury[12] revealed a consensus quality score (QV) of 46.2 (corresponding to a base accuracy of 99.998%) and a completeness rate of 99.4% (Supplementary Table 2). LTR Assembly Index[13] (LAI) evaluation revealed that the Monoe-Viroflay assembly had a very high LAI score (20.32), much higher than that of the Sp75 assembly (1.42) and also higher than that of Spov3 (18.7) and SOL_r1.1 (16.14) (Supplementary Table 3). BUSCO[14] assessment indicated that 97.2% of core conserved single-copy plant genes were found complete in the Monoe-Viroflay assembly (Supplementary Table 3). In addition, around 99.1% of Illumina genomic reads and 98-99% RNA-Seq reads could be mapped to the Monoe-Viroflay assembly. Together these results supported the high base accuracy, completeness and continuity of the Monoe-Viroflay genome assembly.

A total of 624.8 Mb (69.9%) of the Monoe-Viroflay genome assembly (Supplementary Table 4) were repetitive sequences, which was similar to the repeat content in Sp75 (74.4%)[9], Spov3 (69.4%)[10], and SOL_r1.1 (71.2%)[11]. Long terminal repeat (LTR) retrotransposons, mainly in *Copia* and *Gypsy* subfamilies, were the major group of transposable elements in spinach, and their distribution across the chromosomes showed a distinctive pattern between subfamilies (Supplementary Fig. 4). A total of 28,964 protein-coding genes were predicted in the Monoe-Viroflay genome and 98.6% of them were supported either by RNA-Seq reads or by homologs in different protein databases (Fig. 1c and Supplementary Table 5).

### Chromosome evolution and comparative genomics.
Spinach has six monoploid chromosomes, which is less than most, if not all, of the sequenced plants in the order Caryophyllales. To understand how the genome has evolved in spinach, we reconstructed the ancestral karyotype of the family Chenopodiaceae, which included sequenced species sugar beet (*Beta vulgaris*)[15], garden orache (*Atriplex hortensis*)[16], quinoa (*Chenopodium quinoa*)[17], and spinach, using amaranth (*Amaranthus hypochondriacus*)[18] from the family Amaranthaceae as the outgroup. A total of 820 conserved syntenic blocks comprising 7,467-11,992 orthologous genes in sugar beet, garden orache, and amaranth were identified (Fig. 2). Inference on the gene adjacencies reconciled nine protochromosomes or conserved ancestral regions (CARs) consisting of 11,861 protogenes

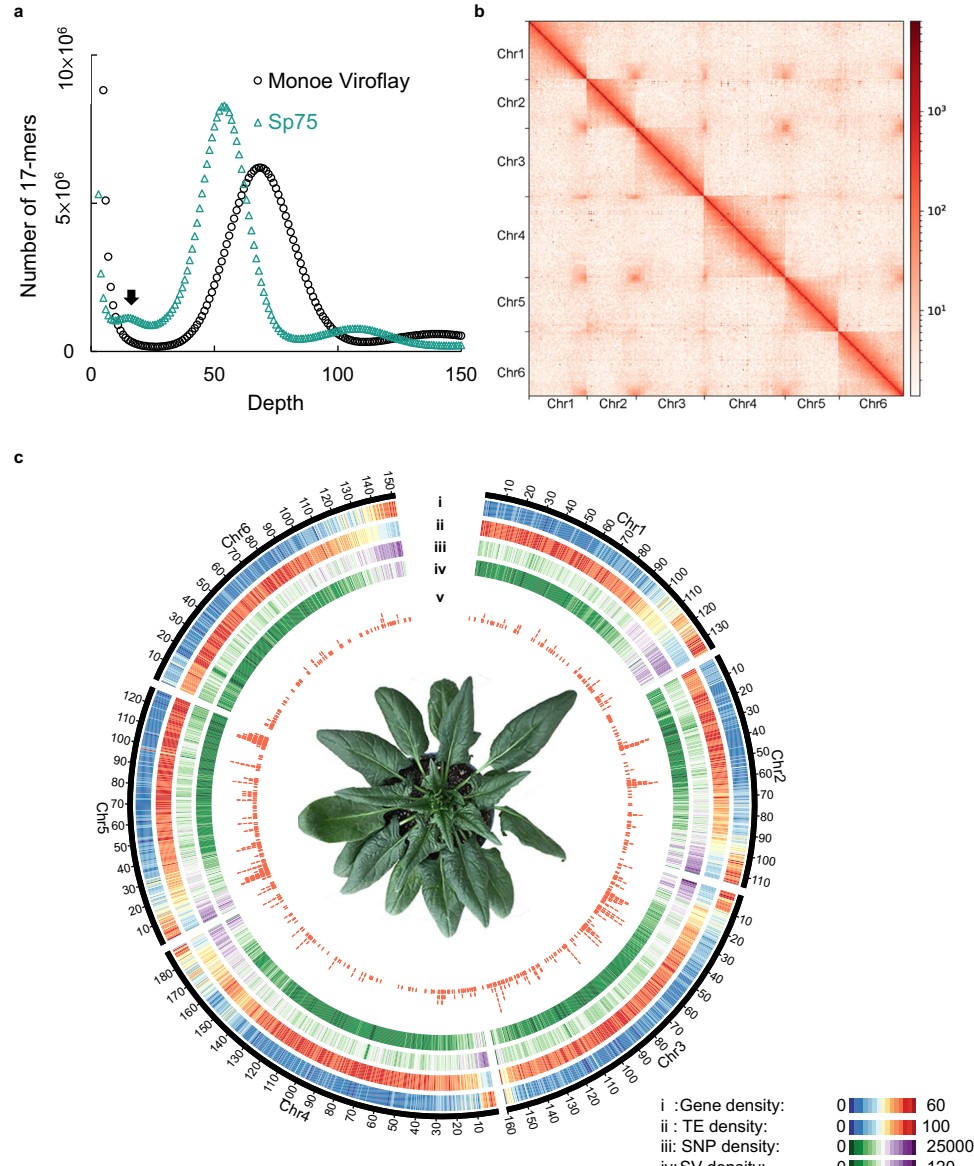

**Fig. 1 Genome of the spinach Monoe-Viroflay. a** 17-mer spectrum of Illumina reads of spinach cultivars Sp75 and Monoe-Viroflay. The main peak represents the homozygous peak and the small peak (pointed by arrow) in Sp75 represents the heterozygous peak. **b** Hi-C interaction heatmap of the assembled Monoe-Viroflay genome. Color bar at the right represents the density of Hi-C interactions, which are indicated by number of links at the 1-Mb resolution. **c** Circos display of the Monoe-Viroflay genomic features. (i) Gene density, (ii) Transposable element (TE) density, (iii) SNP density, (iv) SV density, and (v) Distribution of domestication sweeps across the genome.

in the ancestor of Chenopodiaceae, which were derived from the ancestral eukaryote karyotype (AEK) that has seven proto-chromosomes, followed by a whole-genome triplication occurred ~123 million years ago (a.k.a. the γ event)[19] (Fig. 2). Comparison of AEK and ancestral Chenopodiaceae karyotype (ACK) suggested a substantial genome rearrangement manifested by the different number and architecture of chromosomes (Fig. 2). In contrast, the structure of ACK was overall well maintained in the selected extant species, including the tetraploid quinoa, but not in spinach (Supplementary Fig. 5). We found that all of the Chenopodiaceae proto-chromosomes had undergone fissions and fusions compared with the spinach genome, accompanied with 216 translocations found in spinach, which were remarkably more than those in other species with the same ploidy level, suggesting substantial genome rear-rangements specifically occurred in spinach during the evolution of Chenopodiaceae (Fig. 2). We found that such high level of genome rearrangement correlated with the percentage of repeat sequences

(70% in spinach and 42–66% in other species)[15–18] and might be partly attributed to the recent burst of LTR retrotransposons in the spinach genome[9].

In addition to the reduction of chromosome numbers, there were relatively more gene families under contraction in spinach than in other Chenopodiaceae species (Supplementary Fig. 6 and Supplementary Data 2). These contracted gene families included those responsive to auxin and associated with secondary metabolism such as terpene synthases, whereas genes encoding sugar transporters, F-box domain-containing proteins and transcription factors were apparently expanded (Supplementary Data 3). We predicted 115 NBS-LRR genes in the Monoe-Viroflay genome and around half of them (58) were located in the R gene clusters (Supplementary Data 4 and Supplementary Fig. 7). Compared to closely related Chenopo-diaceae species, the spinach genome encoded fewer R genes, indicating a possible secondary loss of R genes in this species (Supplementary Fig. 8). Plants in the order Caryophyllales are

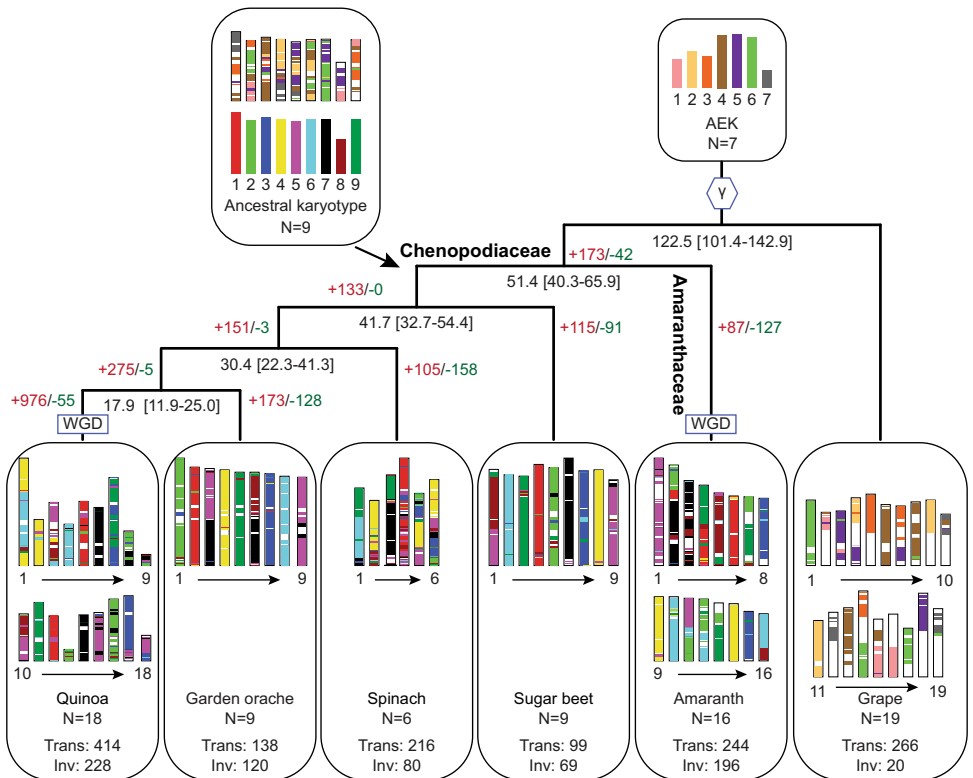

**Fig. 2 Reconstruction of Chenopodiaceae ancestral chromosomes.** Genomes of quinoa (*Chenopodium quinoa*), garden orache (*Atriplex hortensis*), spinach (*Spinacia oleracea*), sugar beet (*Beta vulgaris*), and amaranth (*Amaranthus hypochondriacus*) were used for ancestral genome reconstruction. Chromosomes of the Chenopodiaceae ancestor are painted according to the colors of ancestral eukaryote karyotypes (AEK), while chromosomes of other species are painted following the color palette of the Chenopodiaceae ancestor with nine protochromosomes. Black numbers under the branch of the tree represent the divergence time (million years ago) and the 95% highest posterior density range (in the bracket). Red and green numbers on the tree represent numbers of expanded (+) and contracted (-) gene families across the evolution of the species. Trans: translocation; Inv: inversion.

known to produce betalains, tyrosine-derived pigments that function in the attraction of animal pollinators and dispersers, photoprotection and conferring tolerance to drought and salinity stresses[20]. Consistently, we found that genes participating in betalain biosynthesis were among those expanded in the Caryophyllales (Supplementary Figs. 9 and 10).

**Genome variation map of spinach.** We sequenced genomes of 305 wild and cultivated spinach accessions, including 295 of *S. oleracea* collected worldwide, and all available accessions of *S. tetrandra* (*n* = 3) and *S. turkestanica* (*n* = 7) in the U.S. National Plant Germplasm System (Fig. 3a and Supplementary Data 5). In total, 5.5 Tb Illumina sequences with an average depth of 15.9× and coverage of 98.2% of the assembled Monoe-Viroflay genome were generated. This rich dataset allowed us to identify 17,760,485 single nucleotide polymorphisms (SNPs) and 68,328 structural variants (SVs) in the germplasm collection (Supplementary Table 6). About 49.9% of SNPs were located in or near (within 5 kb up- or downstream of the coding region) the genic region and 2.91% were predicted to have medium or high effects, including changes in the start/stop codons and acceptors/donors in the splicing sites, and non-synonymous changes (Supplementary Table 7). We found that 56.5% SNPs existed among *S. oleracea* and *S. turkestanica* accessions, whereas 97.5% SNPs existed among *S. oleracea* and *S. tetrandra* accessions, suggesting that *S. tetrandra* is genetically more distant to the cultivated spinach than *S. turkestanica*. Such a diverse gene pool of the wild species provides a useful genetic resource for spinach improvement.

**Genetic diversity and population genomics.** Based on the high-density SNP data, the nucleotide diversity (π) of the *S. oleracea* population was estimated to be $1.33 \times 10^{-3}$, which is slightly lower than that of the wild species *S. turkestanica* (π = $1.52 \times 10^{-3}$) (Supplementary Table 8). The estimated genome-wide nucleotide diversity was approximately twice of that estimated based on the transcriptome variants[9], consistent with the purifying selection on the transcribed regions. Estimation of population differentiation ($F_{ST}$) indicated a very low $F_{ST}$ (0.03) between *S. oleracea* and *S. turkestanica*, which is distinguishable from other crops, such as an $F_{ST}$ of 0.36 between cultivated and wild rice[21] and 0.30 between cultivated watermelon and wild *Citrullus mucosospermus*[22], reinforcing a weak bottleneck during the domestication of spinach. Principal component analysis (PCA) based on the SNPs from the fourfold degenerate sites could not distinguish the cultivated *S. oleracea* and the wild *S. turkestanica* accessions due to the abundant genetic variation in *S. tetrandra* (Supplementary Fig. 11), whereas excluding *S. tetrandra*-specific SNPs separated the wild *S. turkestanica* and the *S. oleracea* accessions, as well as the Asian and European subpopulations of *S. oleracea* (Fig. 3b). Model-based clustering and phylogenetic analyses further supported a much closer relationship between *S. oleracea* and *S. turkestanica* than between *S. oleracea* and *S. tetrandra*, and the presence of subgroups in *S. oleracea* that corresponded to their geographic origins (Asia and Europe; Fig. 3c and Supplementary Fig. 12). One cultivated (USX) and two wild (PI 647862 and PI 608712) accessions were placed into suspicious groups, which were later proved to be misclassified by checking their seed phenotype, and were therefore excluded from the population genetic analyses. Assessment on the

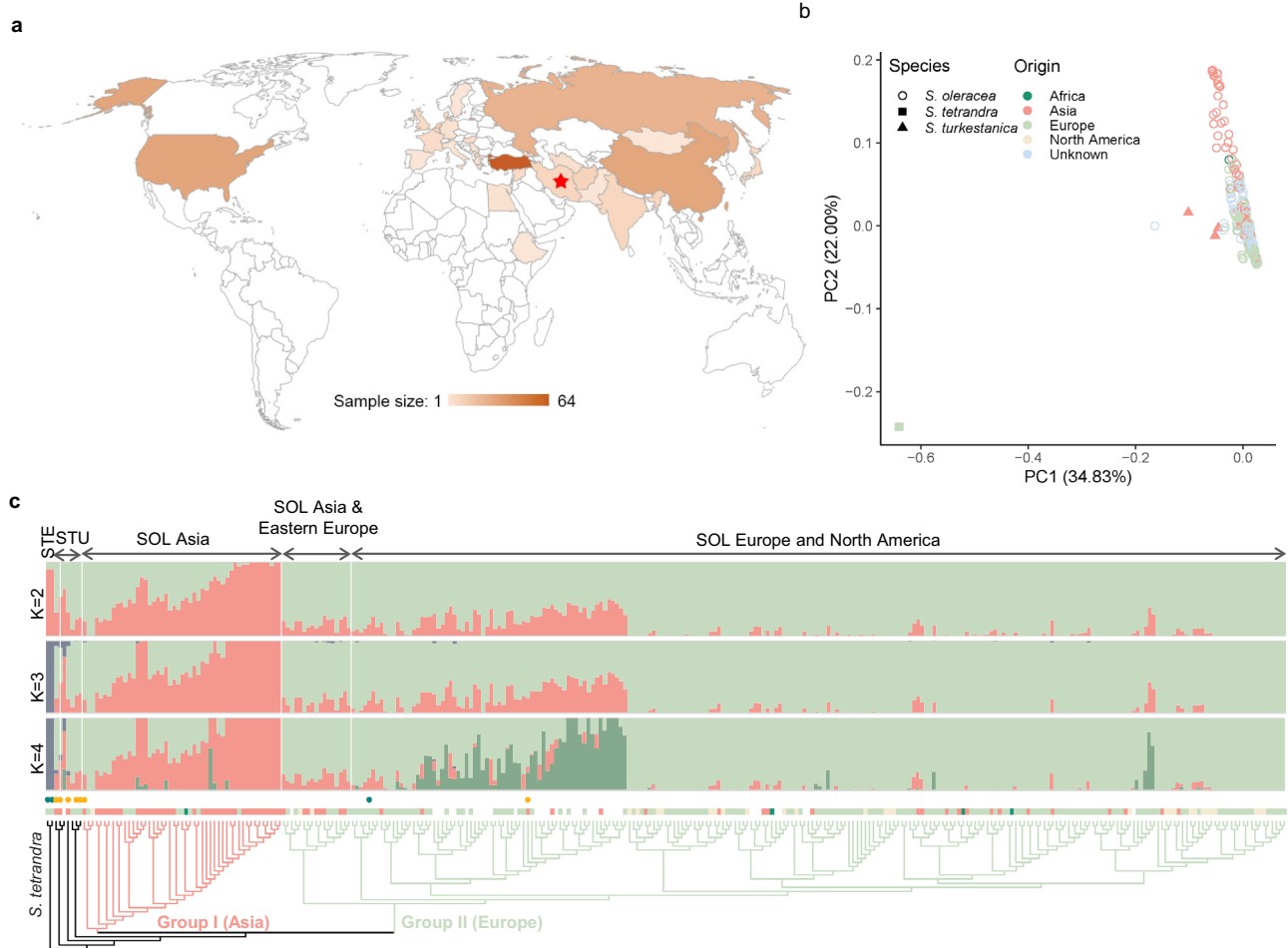

**Fig. 3 Phylogenetic relationships and population structure of *Spinacia* accessions. a** Geographic distribution of the sampled cultivated and wild spinach accessions. Red star indicates the center of origin of spinach. **b** Principal component analysis of spinach accessions using SNPs at fourfold degenerate sites excluding those specific to *S. tetrandra*. **c** Phylogenetic and model-based clustering analyses of spinach accessions.

nucleotide diversity indicated that Asian subpopulation (Group I; $\pi = 1.54 \times 10^{-3}$) of the cultivated spinach was genetically more diverse than the European subpopulation (Group II; $\pi = 1.23 \times 10^{-3}$), reinforcing the origin of spinach in Asia. Moreover, we found that the $F_{ST}$ (0.06) between the Asian and European groups was even higher than that between the cultivated and wild spinaches, suggesting an elevated genetic diversity among geographic populations of cultivated spinach (Supplementary Table 8).

Spinach is an out-crossing species, and the decay distance of linkage disequilibrium (LD) in *S. oleracea* (600 bp at $r^2 = 0.2$; Supplementary Figs. 13, 14) was far shorter than other crops such as rice[21] and tomato[23], but similar to apple[24] that is also an out-crossing species, further supporting the weak bottleneck during spinach domestication. The very rapid LD decay in the cultivated spinach may be useful for identifying the associated QTL intervals that resulted in identification of less spurious candidate genes in GWAS experiments.

**Genome-wide association studies of spinach agronomic traits.** The high-density genomic variation map generated in this study allowed us to dissect genetic architecture of complex spinach traits using GWAS. Twenty agronomic traits, including bolting, flowering, oxalate content, sex expression, downy mildew (DM) resistance, DM incidence, and those associated with plant morphology ($n = 3$), leaf ($n = 3$), and petiole ($n = 8$), were investigated in

303 spinach accessions (Supplementary Data 6). Diverse phenotypic variations were observed for these traits, with the coefficient of variation ranging between 9.76 and 51.20% (Supplementary Table 9). Correlation analysis was carried out on these 20 traits, and the results indicated that plant height, plant width, leaf length, leaf width and petiole length traits were all highly positively correlated ($r$ values of 0.608–0.883), as well as leaf division and leaf shape ($r = 0.668$) and bolting and flowering ($r = 0.808$), while as expected, DM resistance and DM incidence showed a high negative correlation ($r = -0.796$) (Supplementary Data 7).

Based on 5.5 million SNPs with minor allele frequency $\geq 0.05$, we identified 372 significantly associated signals ($\alpha = 0.05$) for 12 traits and 34 associated signals for the other 8 traits (Supplementary Figs. 15–21 and Supplementary Table 10). Improving DM resistance is the main focus in spinach breeding. Using GWAS, we identified a significantly associated region on chromosome 3 (23,583–1,599,751 bp) for both DM resistance and DM incidence, where the lead SNPs for the two traits were located close to each other and coincided with the DM resistance marker *Pfs-1* (ref. [25]) and the fine-mapped DM resistance region[26] (Fig. 4a–c), suggesting the high possibility of this region conferring DM resistance in spinach. The defined region contained 150 genes including six encoding NBS-LRR proteins (grouped into three clusters), five encoding receptor kinases including one MKK7 and one encoding a receptor protein. Most of the known genes conferring DM resistance in Arabidopsis

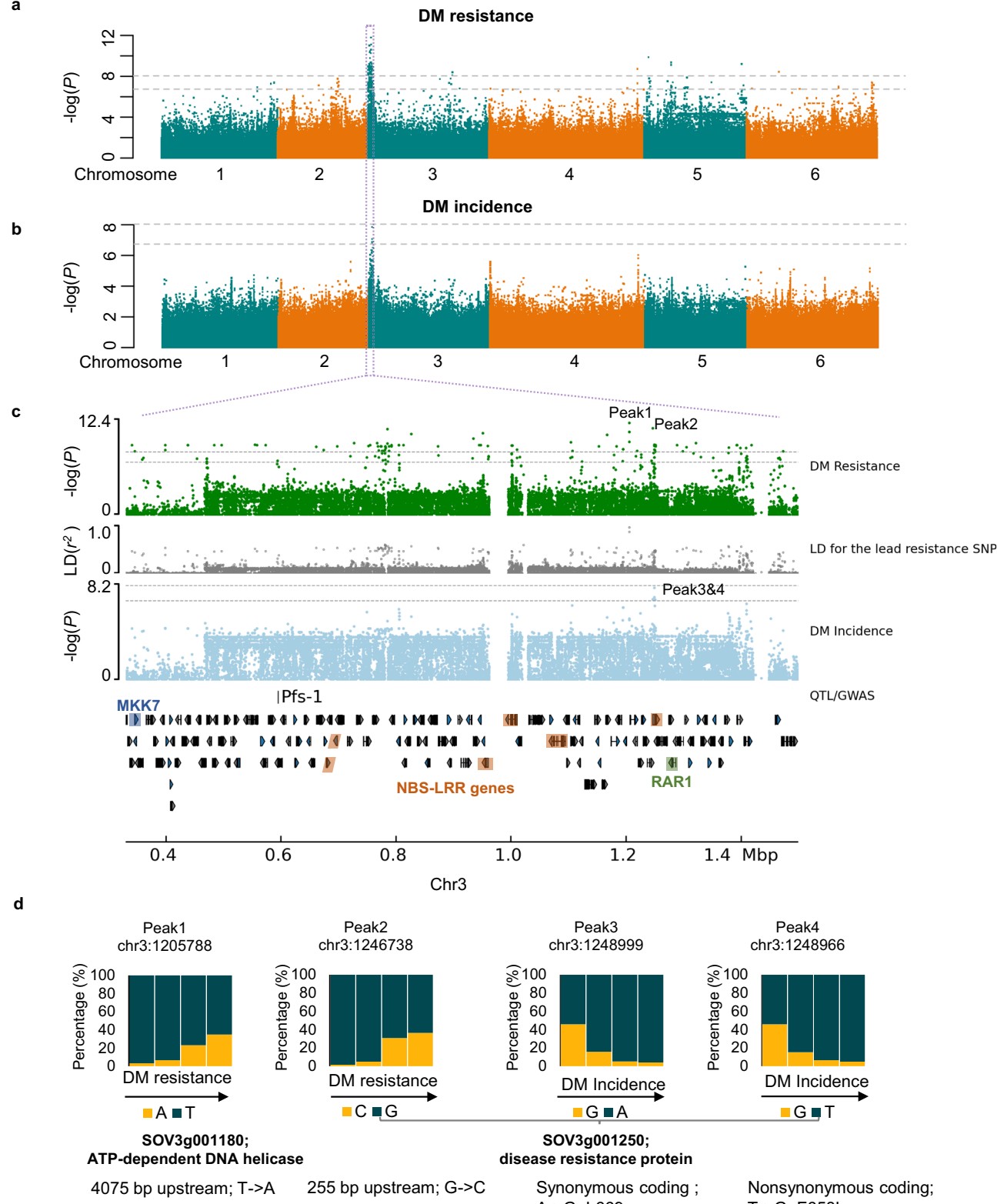

**Fig. 4 GWAS of spinach downy mildew (DM) resistance. a, b** Manhattan plots of GWAS of DM resistance (**a**) and DM incidence (**b**). Gray horizontal dashed lines indicate the Bonferroni-corrected significance thresholds of GWAS ($\alpha = 0.05$ and $\alpha = 1$, respectively). **c** Major associated region with both DM resistance and DM incidence, with genes and the known marker *Pfs*-1 plotted. Gray track indicates the LD ($r^2$) between the lead SNP (Peak1) of DM resistance and other individual SNPs in this region. A kinase receptor gene, *MKK7*, six NBS-LRR genes, and the *RAR1* gene are highlighted in blue, brown, and green, respectively. **d** Allele frequencies of the top two lead SNPs for the DM resistance (Peak1 and Peak2) and DM incidence (Peak3 and Peak4) in the spinach nature population. Source data are provided as a Source Data file.

belong to the NBS-LRR family[27]. Consistently, we found that the lead SNPs were located either upstream or within an NBS-LRR gene, *SOV3g001250* (Fig. 4d). In addition, an orthologous gene of *RAR1*, which is required for NBS-LRR protein accumulation and signaling in Arabidopsis[28], was found in the same region close to *SOV3g001250* (Fig. 4c and Supplementary Data 8). These together suggest that the spinach *Pfs-1* locus might be governed by clustered R genes, with *SOV3g001250* being a potential candidate contributing to DM resistance. Besides the major signal, several minor peaks associated with DM resistance were identified on other chromosomes. The peak SNP on chromosome 4 (Fig. 4a, b and Supplementary Data 9) resided in the promoter of *WSD6* (*SOV4g053700*), which encodes a wax ester synthase that may reinforce the physical barrel of spinach upon pathogen invasion. These data suggest that DM resistance might be multi-faceted in the nature population of spinach.

Spinach is a dioecious plant, which constitutes separate male and female individuals. We identified a genomic region on chromosome 4 (92–103 Mb) showing strong LD and significant association with spinach sex type (Supplementary Fig. 20c). This is consistent with a recent finding that chromosome 4 is a sex chromosome and the male-specific region of the Y chromosome has been maintained through recombination suppression caused by a large inversion located in the middle of chromosome 4 (ref. [29]), which overlaps with our GWAS signals.

Plant type (erect, semi-erect or spreading) is a complex trait that is governed by genes regulating transcription, hormone biosynthesis/signaling and cell cycle[30]. In spinach, we identified numerous regions across the six chromosomes that showed association with plant type. The top 2 signals, which were located on chromosome 2 (~85 Mb) and 3 (~58 Mb) (Supplementary Fig. 15a), respectively, overlapped with one gene having unknown function (*SOV2g021360*) and one encoding a DEAD-box ATP-dependent RNA helicase (*SOV3g025280*) (Supplementary Data 8). In addition, homologs (*SOV4g047390* and *SOV5g010230*) of Arabidopsis *vacuolar pyrophosphatase1* (*AVP1*) and *TEOSINTE BRANCHED1* (*TB1*) were found in other associated regions. *AVP1* and *TB1* have been reported to regulate plant organ development in Arabidopsis, rice, and maize[31–33], and their homologs may also contribute to plant development in spinach.

We searched for genomic component determining the length (height) and width of leaf, petiole and the whole plant, and identified two regions on chromosome 1 with one associated with leaf length, leaf width and plant width (region 1: 61,956,104–62,056,104 bp) and the other with plant height and petiole length (region 2: 72,249,831-72,349,831) (Supplementary Fig. 15, 16 and 19; Supplementary Data 8). Region 1 contained three genes, including one encoding a FAR1-related protein (*SOV1g011880*), whose orthologue in Arabidopsis has been reported to regulate rosette leaf and plant size[34]. In addition, we found another FAR1-related gene (*SOV6g004340*) close to the lead SNP of the petiole width, suggesting the FAR1 gene family as the potential candidates for organ size control in spinach.

A region containing the peak of leaf surface texture was detected near the end of chromosome 6, with a β-tubulin gene (*SOV6g040410*) located in this region (Supplementary Fig. 17a and Supplementary Data 8). It has been reported that microtubule-mediated cell growth anisotropy contributes to leaf flattening[35], which makes this β-tubulin gene a promising candidate for future research on the molecular basis of spinach leaf surface texture. A genomic region significantly associated with leaf base shape was identified on chromosome 3 (~30 Mb). Genes close to the peak were predicted to encode an early-responsive to dehydration stress protein (*SOV3g017870*) as well as a protein of unknown function (*SOV3g017860*) (Supplementary Fig. 18a and Supplementary Data 8). GWAS analyses of bolting and flowering traits were also performed, which yielded several

associated regions across the six chromosomes (Supplementary Fig. 20a, b and Supplementary Data 8), including a region harboring genes encoding MADS-box transcription factors (*SOV6g023690* and *SOV4g008150*), which are known to be involved in regulating flowering time in plants.

Oxalic acid naturally exists in spinach and can chelate easily with $Ca^{2+}$ and $Mg^{2+}$ to form less soluble salts known as oxalates. It is reported that soluble oxalate is the major existing form of oxalate in spinach[36] and when absorbed in the digestive tract can cause a negative impact on human health by reducing mineral absorption and contributing to the formation of kidney stones[37,38]. This makes breeding of spinach cultivar with reduced soluble oxalate content of great interest. We identified a region located on chromosome 5 (915,444-1,022,853 bp) that was associated with the oxalic acid content (Supplementary Fig. 21). The region contained 15 genes (Supplementary Data 8), of which *SOV5g000760* encoded a heavy metal transport/detoxification superfamily protein and *SOV5g000810* encoded a ZIP metal ion transporter family protein. It is known that the increased level of calcium in spinach reduces the soluble oxalate content, and adding calcium ions is recommended when making spinach juice to make it safer for the consumers[36]. We hypothesize that the two transporters in the associated region may regulate the soluble oxalate content in spinach by affecting the calcium and other ion levels in spinach cells. This finding proposes a new approach to breed spinach with reduced soluble oxalates. However, functions of the two transporters and their effects on changed ion levels in spinach need to be further investigated.

**Domestication of spinach.** Human selection on the major crop plants usually results in convergent evolution of morphological traits in seed, root/tuber and fruit, such as enlarged edible parts and reduced undesirable flavor[39]. However, there are exceptions for the leafy vegetables. One example is the *Brassica* vegetables, which show diversifying selection enforced by human[40]. Spinach is another special example of leafy vegetables, which shows indistinguishable leaf/plant morphology among the cultivated and wild species[8], and the wild *S. turkestanica* has been consumed in its natural habitat by the local farmers. The relatively short history of human cultivation[41] and the relatively good quality of wild spinach for human purposes explain why the two species have little divergence at both genetic and morphologic levels. Despite the overall high similarity of morphological traits associated with human selection, there are still some differences that exist in the wild and cultivated species, e.g. the morphology of the pistillate flowers[42], with wild spinaches having clusters of fused flowers and cultivated ones showing clusters of separated flowers, which leads to the spiny aggregate fruit containing many seeds in *S. turkestanica* and detached seeds in *S. oleracea*[8]. To investigate how human selection has altered the genomes of *Spinacia* species, we searched for signatures of selection in the spinach genome, by comparing *S. oleracea* and the closest wild relative *S. turkestanica*. In total, 996 selective sweeps in the *S. oleracea* genome were identified, spanning 17.6 Mb and covering 748 genes (Supplementary Table 11). The sweeps coincided with many known QTLs and GWAS signals identified in this study that were associated with the important traits, including flowering, bolting, plant type, leaf surface texture, leaf base shape, and petiole color and width (Fig. 5a and Supplementary Fig. 22). We found that all the wild *S. turkestanica* spinaches have smooth leaves, while cultivated spinaches in the Asia group grow both smooth and wrinkled leaves and most accessions in the Europe group possess wrinkled leaves (Fig. 5b), indicating that the wrinkled leaf phenotype may be a desirable trait during human selection process. The identified selected region overlapped with the associated

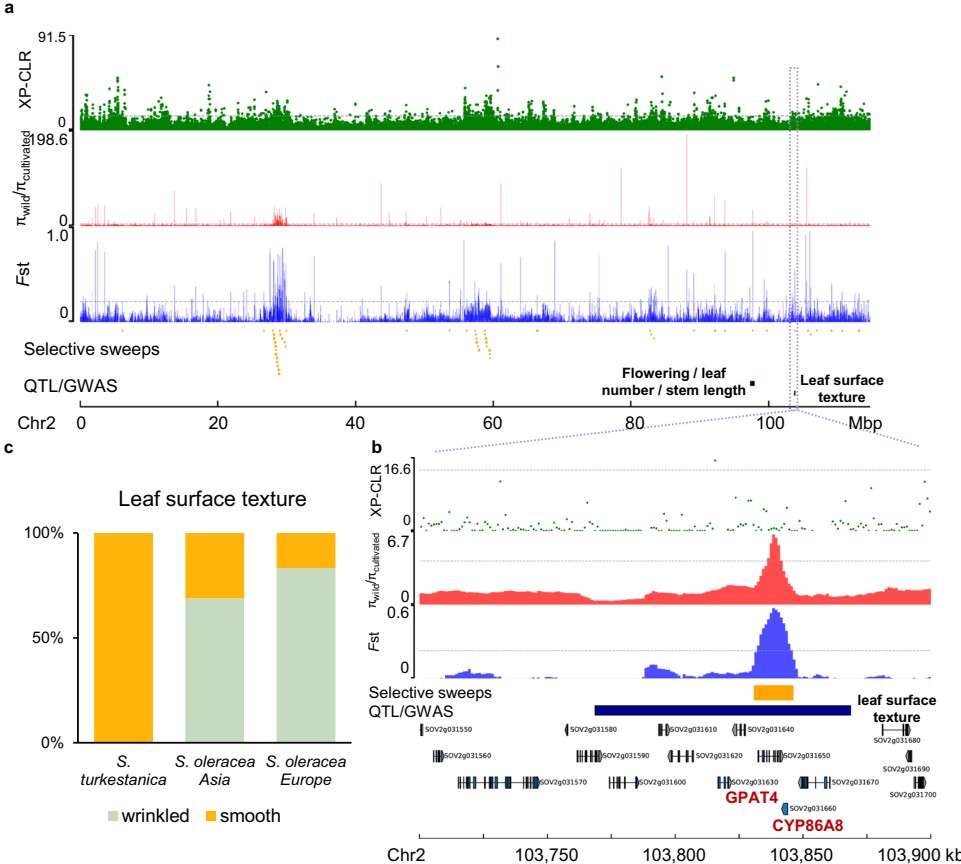

**Fig. 5 Genome-wide screening and functional annotations of domestication sweeps. a** Selective signals on chromosome 2. Gray horizontal dashed lines indicate the top 1% thresholds of the scores/values derived from three approaches, XP-CLR, $F_{ST}$ and nucleotide diversity ($\pi$) ratio. Putative selected regions (orange rectangles) and QTL/GWAS signals overlapping with the identified selective sweeps (black rectangles) are shown below these tracks. GWAS signal of leaf surface texture was identified in this study. Selective signals on other chromosomes are provided in Supplementary Fig. 22. **b** Zoomed-in view of the region containing the GWAS signal of leaf surface texture. **c** Distribution of the leaf surface texture phenotype (wrinkled or smooth) in the three *Spinacia* groups. Source data are provided as a Source Data file.

region of leaf surface texture, which harbored two genes involved in the cutin assembly[43], glycerol-3-phosphate acyltransferase 4 (*GPAT4; SOV2g031630*) and cytochrome P450 (*CYP86A8/LCR; SOV2g031660*) (Supplementary Data 10), unraveling the strong candidate genes regulating the leaf texture trait in spinach.

## Discussion

Heterozygosity and repetitiveness are among major factors limiting assembled genome sequence quality of plants including spinach. Here we assembled a highly accurate, continuous, and complete spinach genome using PacBio long reads and Hi-C interaction maps. The assembled chromosome-scale reference-grade genome is much improved compared with previously published assemblies[9–11], partly due to the use of highly homozygous line "Monoe-Viroflay" and long-read and Hi-C sequencing technologies. The genome provides a valuable resource for future comparative genomic, genetic mapping, and gene cloning studies. The ancestral karyotype of family Chenopodiaceae comprises nine chromosomes; however, spinach only has six, suggesting chromosome reduction possibly mediated by transposable elements[44], which have been remarkably expanded in the spinach genome (Supplementary Table 4).

Marker-assisted selection improves the efficiency of breeding programs by enabling accurate selection for traits of interest using highly reliable molecular markers. In spinach, timing of bolting, oxalate content, seed shape, leaf morphology and resistance to

diseases and pests are the major targets for breeding[8]. Our GWAS analyses on 20 agronomic traits have identified numerous genomic regions associated with these traits, and of these traits several have associated genome regions first identified in this study. Downy mildew is the economically most important disease of spinach affecting commercial spinach production worldwide. We identified a region on chromosome 3 showing strong LD and significant association with DM resistance. This region overlaps with genomic interval identified in previous study[45] and encodes several R genes with one being promising candidate for future functional validation. In addition to the major association signal, minor signals were identified on chromosome 4 and 5, which have not been reported before, suggesting the diversity of DM resistance in the spinach population and providing a great resource for spinach breeding. Consistent with the transcriptome-based analysis, *S. turkestanica* was found to be the possible direct progenitor of cultivated species, despite that transcriptome data underestimate the genome-wide nucleotide diversity of the population. We identified nearly one thousand domestication sweeps harboring 748 genes in cultivated spinach. GWAS signals of flowering, bolting, plant type, leaf surface texture, leaf base shape, and petiole color and width overlap with domestication sweeps, suggesting the role of human selection on phenotypic evolution of spinach.

Our comprehensive genome and population genomic analyses provide insights into spinach genome architecture and evolution, domestication, and genetic basis of important agronomic traits.

The high-quality reference genome and population genomic resources developed in this study are of great value for future biological studies and marker-assisted breeding of spinach.

## Methods

**Plant materials and sequencing.** The Monoe-Viroflay inbred line was provided by Enza Zaden (https://www.enzazaden.com/). The line was derived from a monoecious spinach cultivar Viroflay that was selfed for more than 10 generations, and its genetic homozygosity was confirmed with self-designed markers. The Monoe-Viroflay plants were grown in a greenhouse at Boyce Thompson Institute with a 16 h light (27 °C) and 8 h dark (19 °C) cycle. Young leaves from 20-day-old plants were collected for DNA extraction using the QIAGEN DNeasy Plant Mini Kit following the manufacturer's instructions (QIAGEN, Valencia, CA, USA). DNA quality was determined via agarose gel electrophoresis and the quantity was measured on a NanoDrop (Thermo Fisher Scientific, Waltham, MA, USA). An Illumina paired-end DNA library with an insert size of ~300 bp was constructed using the Genomic DNA Sample Prep kit (Illumina, San Diego, CA, USA) following the manufacturer's instructions, and sequenced on an Illumina HiSeq 4000 platform with paired-end mode. For PacBio sequencing, high molecular weight (HMW) DNA was extracted from young fresh leaves and then randomly sheared to fragments with an average size of 20 kb using g-TUBE (Covaris). The sheared DNA was used to construct the PacBio SMRT library following the standard SMRT bell construction protocol, and the library was sequenced on a PacBio Sequel platform using the 2.0 chemistry (PacBio). Hi-C and Chicago libraries were prepared following the protocols implemented by Dovetail Genomics and sequenced on an Illumina HiSeq 4000 platform.

Transcriptome sequencing was performed for samples collected from young leaf, petiole, stem, root, female flower and fruit tissues. Each sample had two biological replicates. Total RNA was extracted using QIANGEN RNeasy Plant Mini Kit (QIANGEN), and used for strand-specific RNA-Seq library construction following a protocol described in Zhong et al.[46]. RNA-Seq libraries were sequenced on a HiSeq 4000 platform and the reads were processed with Trimmomatic[47] (v0.36) to remove adapter and low-quality sequences, and the processed reads were aligned to the assembled Monoe-Viroflay genome using HISAT2 (ref. [48]) (v2.1).

For genome resequencing, a total of 305 wild and cultivated accessions collected worldwide were used in this study (Supplementary Data 5). DNA isolation from young leaves and Illumina library preparation were performed following the method described above. All libraries were sequenced on an Illumina HiSeq 4000 platform with paired-end mode (2 × 150 bp).

**De novo assembly of the Monoe-Viroflay genome.** PacBio reads were error corrected and assembled into contigs using CANU[49] (v1.7.1) with default parameters except that 'OvlMerThreshold' and 'corOutCoverage' were set to 500 and 200, respectively. The Arrow program implemented in SMRT-link-5.1 (PacBio) was used to correct the draft assembly with long reads and the corrected assembly was subjected to a second-round error correction using Pilon[50] (v1.22; parameters '-fix bases -diploid') with the cleaned Illumina paired-end reads. Subsequently, the error-corrected contigs were aligned against the NCBI nt database to search for putative contaminations, and those with ≥ 90% of the sequences similar to the organelle genomes or microbial genomes were discarded. To scaffold the contigs, Illumina reads from Hi-C and Chicago libraries were processed with Trimmomatic[47] (v0.36) to remove adaptors and low-quality sequences. The cleaned reads were then used for scaffolding with the HiRise program (Dovetail Genomics). The final assembly was manually checked and curated based on the alignments of Hi-C reads and the genetic maps[9].

**Repeat sequence annotation and gene prediction.** The Monoe-Viroflay genome was scanned for miniature inverted-repeat transposable elements (MITEs) and long terminal repeat retrotransposons (LTR-RTs) using MITE-Hunter[51] (v11-2011) and LTRharvest[52] (v1.5.10), respectively. The genome was masked using RepeatMasker (v4.0.8; http://www.repeatmasker.org/) with identified MITE and LTR-RT sequences, and the unmasked portion of the genome was extracted and fed to RepeatModeler (v1.0.11; http://www.repeatmasker.org/RepeatModeler.html) for de novo repeat library construction. Repeats from above were combined to build a final repeat library of the Monoe-Viroflay genome and used to screen the Monoe-Viroflay genome for repetitive sequences using RepeatMasker.

Protein-coding genes were predicted from the repeat-masked Monoe-Viroflay genome with the MAKER-P program[53] (v2.31.10), which integrates evidence from protein homology, transcriptome and ab initio predictions. SNAP[54] (v2006-07-28), AUGUSTUS[55] (v3.3), and GeneMark-ES[56] (v4.35) were used for ab initio gene predictions. Protein sequences from quinoa, sugar beet, the previously assembled Sp75, and the Arabidopsis genomes were included in the annotation pipeline as the homology evidence. RNA-Seq data generated in this study were cleaned with Trimmomatic[47] and assembled using StringTie[57] (v1.3.3b). The complete coding sequences (CDS) were predicted from the assembled transcripts using the PASA pipeline[58] (v2.3.3) and used as transcript evidence for gene prediction.

**Ancestral chromosome reconstruction and gene family evolution.** The ancestral chromosomes of Chenopodiaceae were reconstructed based on the method described in Murat et al.[59]. Briefly, genomes of sugar beet, garden orache and amaranth were used as seed genomes to reconstruct the ancestral Chenopodiaceae karyotype with amaranth used as the outgroup. Genomic synteny between each pair of the three species was identified using MCScanX[60] and the mean Ks of each syntenic block was calculated and used to filter out paralogous syntenic blocks within the amaranth genome (i.e., only the blocks with lowest mean Ks values were preserved). The pairwise syntenic blocks were then combined to build multi-genome blocks, which were used by the MGRA program[61] (v2.3.0 beta) to infer the ancestral karyotype of the selected species.

To infer gene family evolution, orthologous groups among selected species were built by OrthoMCL[62] (v2.0.9) with parameters "E-value < 1e–5; inflation value 1.5". Single-copy orthologous groups were used to build the species phylogeny. Protein sequences of each orthologous group was aligned separately with MAFFT[63] (v7.313) and gaps in the alignment were trimmed with trimAl[64] (v1.2). A maximum likelihood phylogeny was inferred by IQ-TREE[65] (v1.6.7) with concatenated alignments and the best-fitting model, and with 1000 bootstrap replicates. Molecular dating was carried out by MCMCTree in the PAML package[66] (v4.8). The divergence time of monocots-eudicots (150-120 million years ago) was used as a calibration point according to Morris et al.[67]. Modeling of gene family size was performed using CAFE[68] (v 4.2) and the gene birth and death rate was estimated with orthologous groups that were conserved in all selected species.

**Variant calling.** Raw reads of 305 cultivated and wild spinach accessions were processed with Trimmomatic[47] to remove adaptors and low-quality bases, and the cleaned reads were mapped to the Monoe-Viroflay genome using BWA-MEM[69] (v0.7.17). Alignments with mapping quality > 20 were retained for variant calling using GATK[70] (v4.1). The GATK recommended hard filters for the SNP data (QD < 2.0, QUAL < 30.0, SOR > 3.0, FS > 60.0, MQ < 40.0, MQRankSum < −12.5, ReadPosRankSum < −8.0) were applied to remove low-confidence variants. The resulted bi-allelic SNP sites with a missing rate less than 0.5 and quality value greater than 60 were kept for the downstream analyses. SVs were detected using the smoove pipeline (https://github.com/brentp/smoove), which integrates the SV detecting tool LUMPY[71] (v0.2.13) and genotyping tool SVTyper[72] (v0.7.1) and other post-filtration steps, by aligning the Illumina paired-end reads against the Monoe-Viroflay genome.

**Phylogeny and population structure of spinach.** A total of 549,814 bi-allelic SNPs (MAF ≥ 0.02, missing rate < =0.1 and minimum distance between two SNPs ≥ 1 kb) were used for phylogenic analysis. The p-distance matrix of the 305 spinach accessions was calculated with VCF2Dis (https://github.com/BGI-shenzhen/VCF2Dis), and used to build the neighbor-joining phylogeny. Principal component analysis (PCA) was performed with EIGENSOFT[73] (v7.2.1) and the population structure was analyzed using the STRUCTURE program[74] (v2.3.4), both of which used SNPs at the fourfold degenerate sites with S. tetrandra-specific ones excluded. The likelihood of ancestral kinships (K) from 1 to 15 was estimated using 20,000 randomly selected SNPs, and each K was run 20 times. ΔK indicating the rate of change in the log probability of data between successive K values was calculated, and the number of kinships with the highest ΔK suggesting the most likely number of clusters in the population was determined. Linkage disequilibrium (LD) decay was measured by calculating correlation coefficients ($r^2$) for all pairs of SNPs within 500 kb using PopLDdecay[75] (v3.41).

**Detection of domestication sweeps.** Genome-wide scan of selective sweeps was performed by comparing allele frequency between cultivated and wild (S. turkestanica) spinach populations using three approaches, calculating XP-CLR scores[76] with the parameters '-w 1 0.005 200 1000 -p0 0.95', $F_{ST}$, and nucleotide diversity (π) ratios using VCFtools[77] (v0.1.16). $F_{ST}$ and π ratios were calculated based on a sliding window of 10 kb and a step size of 1 kb. Regions ranked top 1% of the scores/values in any two of the methods were defined as putative selective sweeps.

**Planting and phenotyping.** All spinach accessions were planted in the field at the Development and Collaborative Innovation Center of Plant Germplasm Resources, Fengxian Campus of Shanghai Normal University (latitude, 30° 50′ N; longitude, 121° 31′ E) in the winter of 2019, using a randomized complete block design with three replicates, and 10 plants were selected for the phenotyping of agronomic and disease resistance traits for each accession.

Three traits related to plant architecture, plant type, plant height, and plant width, were measured at 35 days after germination. Specifically, the plant type trait was classified into three categories, erect (angle > 60°), semi-erect (45° < angle < 60°) and spreading (angle < 45°), according to the angle between the extension direction of the outer spinach leaf base and the ground plane. The plant height and width were recorded based on the vertical distance from the highest point of the plant to the ground and the maximum diameter of the horizontal projection of the leaf curtain of the plant in its natural state, respectively. Eight leaf traits, including leaf length, leaf width, leaf number, leaf surface texture, leaf shape, leaf apex shape, leaf base shape and leaf division shape, were also recorded at 35 days after germination. The largest leaves of three individuals were selected and measured for

the length and width traits, and the number of unfolded leaves with a length greater than 2 cm was recorded for the leaf number trait. The leaf surface texture trait was recorded as smooth and wrinkled. Leaf shape was categorized into nearly orbicular, ovate, elliptic and halberd. Leaf apex shape and leaf base shape were classified into acute and non-acute, and plane and non-plane, respectively. The degree of leaf division was divided into none, shallow, medium, and deep. Petiole-related traits including petiole length and petiole width were recorded along with the leaf length and width, and the petiole color was recorded as purple and green. Bolting and flowering were recorded as the number of days needed for more than half of the plants having a stem longer than 5 cm and flowering, respectively. Sex type was recorded as male, female or monoecious at the flowering stage.

To measure the soluble oxalate content, 0.1 g spinach leaf with 3.5 ml of deionized water was homogenized for 3 min in a 5-ml centrifuge tube, put in a water bath at 75 °C for 40 min, and then centrifuged at 12000 rpm for 5 min. Around 0.25 ml supernatant was taken, mixed with 0.5 ml $Fe^{3+}$ buffer, 5 ml KCI buffer, 0.3 ml sulfosalicylic acid, and 4 ml distilled water, and sit for 30 min. The absorbance at 510 nm with distilled water as the reference solution was measured. It has been demonstrated that there is a strong linear correlation between oxalic acid concentrations and color degrees[5], therefore the oxalic acid content of spinach leaf could be calculated according to the standard curve.

The resistance and incidence of downy mildew in spinach were determined by artificial inoculation of *Peronospora farinosa* f. sp. *spinaciae* on the spinach seedlings. Specifically, the pathogenic spores were evenly sprayed on the true leaf of the seedlings which were then kept in the greenhouse at a temperature of 20 °C and humidity of above 80%. The resistance and incidence of downy mildew were investigated 10 days after the inoculation. The incidence of each accession was rated based on the number of susceptible plants divided by the number of all plants. The resistance of each accession was characterized as follows:

First, the disease degree of each plant was classified according to the percentage of leaves showing the symptom, 0, 0% (no symptom); 1, 1–25%; 2, 26–50%; 3, 51–75%; and 4, ≥76%. The disease index (DI) was then calculated using the following formula:

$$DI = \left( \sum (s_i n_i) \right) / 4N \times 100 \qquad (1)$$

where $s_i$ is the disease degree, $n_i$ is the number of plants showing this degree of disease and $N$ is the total number of plants surveyed. Finally, the resistance of spinach accession to downy mildew was divided into five levels according to the calculated disease index: high resistance (DI < 1), resistance (1 ≤ DI < 10), medium resistance (10 ≤ DI < 20), susceptible (20 ≤ DI < 45) and highly susceptible (DI ≥ 45).

**GWAS**. A total of 5,511,663 SNPs with a minor allele frequency of 0.01 or greater and a missing data rate of 50% or less in the entire population were used for GWAS. The Balding–Nichols kinship matrix generated with the EMMAX program[78] (v20120210) was used to correct the population structure. GWAS analyses were performed using the linear mixed model implemented in the EMMAX program. The modified Bonferroni correction was used to determine the genome-wide significance thresholds of the GWAS, based on nominal levels of $\alpha = 0.05$ and $\alpha = 1$, corresponding to raw $P$ values of $9.09 \times 10^{-9}$ at $\alpha = 0.05$ and $1.82 \times 10^{-7}$ at $\alpha = 1$, or $-\log_{10}(P)$ values of 8.04 and 6.74, respectively. The associated regions for each trait were defined by extending 50-kb upstream and downstream of the identified SNP markers, and overlapped regions were concatenated using bedtools[79] (v2.26.0).

**Reporting summary**. Further information on research design is available in the Nature Research Reporting Summary linked to this article.

## Data availability

This Whole Genome Shotgun project has been deposited at DDBJ/ENA/GenBank under the accession WWFT00000000 The version described in this paper is version WWFT01000000. Raw genome and transcriptome sequencing reads have been deposited in the NCBI BioProject database under the accession number PRJNA598728 The Monoe-Viroflay genome assembly and annotated genes [http://spinachbase.org/ftp/genome/Monoe-Viroflay/] and SNPs and SVs [http://spinachbase.org/ftp/variant/genome/] in VCF file format are also available at SpinachBase[80] (http://spinachbase.org/). Source data are provided with this paper.

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

## Acknowledgements

The authors thank Drs. Jan Dijkstra and Faira Suidgeest from Enza Zaden for providing seeds and confirming the genetic homozygosity of Monoe-Viroflay. This research was supported by grants from Science and Technology Commission of Shanghai Municipality, China (19070502600 to Quanhua Wang, 19391900700 to X.C., and 18DZ2260500 to Quanhua Wang), the Development and Collaborative Innovation Center of Shanghai (No. ZF1205 to Quanxi Wang), Shanghai Engineering Research Center of Plant Germplasm Resources (17DZ2252700 to Quanxi Wang), Shanghai Municipal Agricultural Commission (No. 2019-02-08-00-02-F01105 to Quanhua Wang), the Open Project of Qinghai Key Laboratory of Vegetable Genetics and Physiology (Grant No. 2018-ZJ-Y18 to X.C.), and the US National Science Foundation (IOS-1855585 to Z.F.).

## Author contributions

Quanhua Wang, C.J., Z.F., Quanxi Wang, and Z.Z. designed and managed the project. X.C., C.X., X.W., C.G., Quanhua Wang contributed to sample collection and phenotyping. C.J. performed genome assembly and annotation. C.J., X.S., and H.S. conducted genomic, population genomic, and GWAS analyses. C.J., X.S., and X.C. wrote the manuscript. Z.F. revised the manuscript. All authors read and approved the manuscript.

## Competing interests

The authors declare no competing interests.
