## [Peer Review File · Nature Communications]

Genomic analyses provide insights into spinach domestication and the genetic basis of agronomic traitsReviewers' Comments:

Reviewer #1:

Remarks to the Author:

The authors build a high quality chromosome-scale genome assembly of a highly homozygous individual of a monoecious spinach cultivar and use it to 1) reconstruct an ancestral estimation of the Chenopodiaceae genome, 2) resequence a diversity of male and female germplasm of 3 species, 3) phenotype the collection and perform GWAS for a variety of traits, and 4) estimate signatures of putative domestication sweeps. Overall I think the manuscript does an impressive job of integrating a large amount of genomic data with careful phenotyping. The assembly is high quality, the resequencing dataset is impressive, and the phenotyping data is excellent. The QTLs and genes described are mostly descriptive, which is expected and reasonable since transformation and CRISPR is not widespread in spinach. I have several major and minor comments for the genome, largely relating to the lack of discussion or integration of the known XY sex chromosome pair into major analyses.

Major comments:

1. The sex chromosome is briefly mentioned in the manuscript, but it seems especially important to describe the complications of using a monoecious reference genome to study a dioecious species with a known sex chromosome pair. In this sense, the reference genome is incomplete for the species. The genome accession is a stable monoecious individual, but the resequenced accessions are either male, female, or unknown. Is the monoecious Viroflay-derived reference genome individual a derivative of an XY male individual, or an XX female individual, or something else (YY)? If chr4 is derived from a Y chromosome, one would expect to see the male-specific mapping in some regions of the non-recombining region in addition to 50% read coverage relative to the homologous X chromosome and autosomes. Depending on which sex chromosome chr4 is derived from, the X or Y, there could be quite a few genes that are impossible to map reads to since they are not found in the assembly (e.g. Y-specific region). Supplementary figure S17 suggests that the non-recombining region of the sex chromosome is quite large — perhaps 15% of chromosome 4 length. I was expecting the signal (high TE density, potentially lower gene content) from such a sex chromosome to be more obvious on Figure 1A, which is surprising. Phenotypic sex is missing from quite a few individuals; this could easily be inferred from the data (e.g. published Y-specific markers) and would be valuable for the community and the manuscripts GWAS analyses, in addition to testing alternative hypotheses for some major conclusions where sex could be a confounding factor (see point 4).
2. Further, since non-recombining regions of Y chromosomes are present at 1/2 coverage relative to an X, any coverage-based mapping quality cutoffs would possibly eliminate variants from being called. This could be a problem at ~15X haploid coverage averages (7.5X coverage for the Y chromosome).
3. The ancestral state reconstruction is interesting, but it is unclear to me the purpose of this analysis for the manuscript since it is not integrated into the story. Substantial genome rearrangements specific to Spinach were found, compared to the ancestral karyotype; do these blocks overlap with any of the QTLs or domestication loci found?
4. *Turkestanica* has previously been described as the closest relative to oleracea (Hammer K 2001). In the current analysis, the sample size of *Turkestanica* is quite small (n=3), and it appears that at least two of the accessions are collected from the same physical location or population in Georgia. Compared to the hundreds of oleracea accessions, it seems that the sample sizes of the other two species is too small to make conclusions based on a PCA. Going back to the sex chromosome question, it is possible that a divergent sex chromosome is driving the PCA patterns for *turkestanica*: are these 3 *turkestanica* individuals the same sex, or a mix of males and females (the supplemental data shows "NA" for sex)? If all are the same sex, it is worth testing if sex is driving these PCA patterns, and would benefit from the suggestion in comment #1 to genotypically infer the sex of all

resequenced individuals with known sex markers.

5. The supplemental information is incredibly clear and easy to understand. Thank you for taking the time to present it so well.

Minor comments:

L44: basal is not the right word here, as it implies a hierarchy to phylogeny. Terms like “basal” reflect misconceptions about evolution and phylogeny.

L74: 15X coverage is not “deep sequencing”.

L180: Should say n=7 I believe

L184: I would consider this a “germplasm collection” rather than population.

L189: Please clarify this sentence, e.g. Does this mean that 56.5% of the total SNPs called are shared between oleraceae and turkestanica?

L206: Are all three species able to inter-cross? Is there ongoing gene flow?

L223: Define population fixation index here.

L277: Plant type/form is introduced in the manuscript without any description. What is type/form?

L372: I would reserve the word “cutting-edge” for the latest technology, which I do not believe this assembly qualifies for. The manuscript is unclear on what PacBio platform was used, but it does not appear to be HiFi.

L380: Please clarify this sentence. What selective force is acting upon the sex chromosome during the evolution of Chenopodiaceae? Are other species in the family dioecious? Is there a shared origin of the sex chromosome with other species in the family?

L422: What PacBio sequencing platform was used? CLR? Which machine — RSII, Sequel-I, Sequel-II?

Reviewer #2:

Remarks to the Author:

The authors did excellent job for the research and for the manuscript! From the manuscript, authors create a new chromosome-scale reference genome assembly of spinach-inbred line ‘Monoe-Viroflay’ with a total size of 894.3 Mb and 98.3% anchored and ordered on the six chromosomes. The new spinach ‘Monoe-Viroflay’ genome assembly reference has higher genome coverage than those published in public, as I know it is the highest coverage so far. Authors conducted genome-wide association studies of 20 agronomical traits and identified numerous significantly associated regions and candidate genes for the 20 traits, which will be useful in spinach molecular breeding. I recommend this manuscript to be accepted and published in Nature Communications.

Comments are

1. For the section “Planting and phenotyping” from lines 526 to 573, it will be nice to add some detail for phenotyping, such as how many replicates was used for the randomized complete block design (RCBD) design? A total of plants were used to do phenotyping for each trait or how many plants were used in each replicate.

2. It will be nice to do some statistics analysis (such as ANOVA, distribution, correlation) for those quantitative phenotypic data such as plant height, plant width, leaf length, leaf width, leaf number, oxalate content, and downy mildew (DM) incidence and disease index (DI).

3. For Fig. 3 of line 192-193 on page 9, the two-dimension PCA plot seems there is only one cluster (group) and almost all spinach genotypes (lines) merge at one place (3b) and the similar figure of the Supplementary Fig. 9 seems something wrong. For the 3c, it will be nice to add the cluster (sub-population) information from Fig 3c to the Table S10 for each of the 305 spinach accessions when K=2, 3, and 4 in order for readers to see each spinach accession belongs to which cluster (group) easily.

4. For Supplementary Fig. 11 "Linkage disequilibrium (LD) decay pattern of cultivated spinach (*Spinacia oleracea*)", it will be nice to have six LD decay patterns by each chromosome, respectively!

Reviewer #3:

Remarks to the Author:

Cai, Sun, Xu et al. present a comprehensive manuscript regarding spinach genome, evolution, domestication and genetic basis of agronomic traits. They built a significantly improved spinach reference genome using state-of-the-art technologies which represents an important resource for applied and basic plant research. Using this resource, they notably performed macro-evolutionary analyses at the karyotype level. They resequenced 305 cultivated and wild spinach accessions and provided insight into spinach genetic diversity and population differentiation, performing GWAS of 20 agronomical traits and identifying domestication sweeps in the spinach genome. Globally, this study provides valuable resources for facilitating spinach breeding as well as insights into spinach evolution and domestication.

Overall, I find this work relevant and well-conducted. The results presented here will be of immediate interest to people working on different research areas (evolutionary genomics, plant breeding research, population genomics...).

I have several requests/suggestions for clarification and improvement of the results.

-It looks clear that the spinach reference genome generated here is of high quality and is improved compared to the previous ones (Sp75, Spov3 and SOL_r1.1). However, the alignment of newly generated pseudomolecules against pseudomolecules previously generated would be appreciated. Indeed, the scientific community working with those previous versions would strongly benefit from that. A dot-plot representation like in Supplementary Fig.3 would give a relevant overview.

-As a complement of Fig.1 b (in Supplementary Figures for example), it would be interesting to have plots showing the real numbers regarding gene, TE, SNP and SV densities (scatter or density plot for example). Indeed, heatmaps give a nice overview but are not precise for more advanced observations. Also, the chromosome painting (based on the ancestral karyotype) could be represented under the x-axis of those plots. This would provide direct visualization of the potential correlation between chromosomal rearrangements and gene, TE, SNP, and SV densities.

-Along the same line of the previous comment, it would be interesting to check whether fission and fusion points show particular gene or TE patterns. For example, spinach chromosome 4 (protochromosome 1) which has been shown to be very stable during Chenopodiaceae evolution seems to exhibit high gene density in the two subtelomeric regions, which does not seem to be the case for the other chromosomes. One could wonder how gene and TE densities are conserved across species depending on rearrangements. There is maybe something interesting to dig into in this context.

-In Fig.2, it would be relevant to represent Quinoa and Amaranth WGDs as the gamma triplication is represented. Indeed, those additional and specific WGDs directly explain the chromosome number expansion for those species.

-It is difficult and not accurate to assess it by eyes on the figures, but I am wondering whether the most likely ancestral karyotype is composed of 9 or 10 chromosomes. I am wondering whether the beginning of the Sugar beet chromosome 1 corresponding to the end of the garden orache chromosome 9 should be an independent ancestral chromosome or combined with the other purple regions. Could you please check it out and provide pieces of evidence?

-line 379: You probably mean Supplementary Fig. 3, isn't it?

Note: Line numbers indicated in this response letter are based on the converted pdf file, which are different from those shown in the original Word file.

Reviewer #1:

The authors build a high quality chromosome-scale genome assembly of a highly homozygous individual of a monoecious spinach cultivar and use it to 1) reconstruct an ancestral estimation of the Chenopodiaceae genome, 2) resequence a diversity of male and female germplasm of 3 species, 3) phenotype the collection and perform GWAS for a variety of traits, and 4) estimate signatures of putative domestication sweeps. Overall I think the manuscript does an impressive job of integrating a large amount of genomic data with careful phenotyping. The assembly is high quality, the resequencing dataset is impressive, and the phenotyping data is excellent. The QTLs and genes described are mostly descriptive, which is expected and reasonable since transformation and CRISPR is not widespread in spinach. I have several major and minor comments for the genome, largely relating to the lack of discussion or integration of the known XY sex chromosome pair into major analyses.

Major comments:

1. The sex chromosome is briefly mentioned in the manuscript, but it seems especially important to describe the complications of using a monoecious reference genome to study a dioecious species with a known sex chromosome pair. In this sense, the reference genome is incomplete for the species. The genome accession is a stable monoecious individual, but the resequenced accessions are either male, female, or unknown. Is the monoecious Viroflay-derived reference genome individual a derivative of an XY male individual, or an XX female individual, or something else (YY?)? If chr4 is derived from a Y chromosome, one would expect to see the male-specific mapping in some regions of the non-recombining region in addition to 50% read coverage relative to the homologous X chromosome and autosomes. Depending on which sex chromosome chr4 is derived from, the X or Y, there could be quite a few genes that are impossible to map reads to since they are not found in the assembly (e.g., Y-specific region). Supplementary figure S17 suggests that the non-recombining region of the sex chromosome is quite large - perhaps 15% of chromosome 4 length. I was expecting the signal (high TE density, potentially lower gene content) from such a sex chromosome to be more obvious on Figure 1A, which is surprising. Phenotypic sex is missing from quite a few individuals; this could easily be inferred from the data (e.g., published Y-specific markers) and would be valuable for the community and the manuscripts GWAS analyses, in addition to testing alternative hypotheses for some major conclusions where sex could be a confounding factor (see point 4).

Response: We thank the reviewer for the suggestion. The reference line is a derivative of an XX female individual. This is further supported by the alignments of sex-specific markers (T11A, V20A, SpoX and X12; Wadlington and Ming, 2018, *Theor. Appl. Genet.* 131:1987; Kudoh et al., 2018, *Mol Genet Genomics* 293, 557; Yu et al., 2021, *Plant Reprod*) to the assembled ‘Monoe-Viroflay’ genome. We have added this information in the revised manuscript (Line 82-83).

According to She et al. (2020; bioRxiv; <https://doi.org/10.1101/2020.11.23.393710>), the male-specific region of the Y chromosome is around 10 Mb in size, representing ~1% of the spinach genome. The region harbors 210 protein-coding genes, of which only 13 that do not share any homologous sequences with the X chromosome or autosome. Therefore, the information missing in or not captured by the high-quality ‘Monoe-Viroflay’ reference genome would be minimum. It is worth noting that any single reference genome is incomplete in terms of capturing the entire genetic elements in a species or genus. Therefore, a

pan-genome constructed from genomes of multiple representative accessions would be needed for spinach in the near future; however, we believe this is beyond the scope of this study.

In this study, we collected sex phenotype data for 192 accessions (two-thirds of all accessions). We believe this data is sufficient for a robust GWAS analysis. As for inferencing the sex type of the remaining 1/3 accessions, since we don't have a genome assembly for every accession, we cannot directly search the markers (to infer the expected fragment size) or the male-specific sequences against its genome. In addition, we found that it is not feasible to predict the sex type just based on the read coverage of the male-specific sequences, due to its repetitive nature that often causes very uneven read distribution for that region. While we agree with the reviewer that sex type information for the remaining ~100 accessions would be valuable, we are sorry that we can't confidently infer the sex type based on the available information and data.

2. Further, since non-recombining regions of Y chromosomes are present at 1/2 coverage relative to an X, any coverage-based mapping quality cutoffs would possibly eliminate variants from being called. This could be a problem at ~15X haploid coverage averages (7.5X coverage for the Y chromosome). **Response:** As mentioned above, the male-specific region of the Y chromosome, representing 1% of the genome, was not available in the analysis. In addition, this region mainly consists of transposable elements that are mostly unmappable. Nonetheless, in our SNP detection, we only require sites covered by two or more reads as recommended by the GATK pipeline ("QD < 2.0"; Line 460); therefore, 7.5× coverage is still sufficient for SNP identification in this study.

3. The ancestral state reconstruction is interesting, but it is unclear to me the purpose of this analysis for the manuscript since it is not integrated into the story. Substantial genome rearrangements specific to Spinach were found, compared to the ancestral karyotype; do these blocks overlap with any of the QTLs or domestication loci found?

Response: The main purpose of this analysis is to gain insights into the history of the genome evolution. Spinach has a chromosome number (n=6) remarkably different from closely related species in the Chenopodiaceae, which has a base number of 9, and we were interested in understanding how evolution has shaped chromosome rearrangement during the diversification of Chenopodiaceae. We note that such genome rearrangements in spinach are the consequence of long-term evolution and have nothing to do with domestication and QTLs. It is also worth noting that this type of analysis has been reported in numerous plant genome papers. Here are several examples:

Myburg et al., 2014; <https://www.nature.com/articles/nature13308>

Badouin et al., 2017; <https://www.nature.com/articles/nature22380>

Kreplak et al., 2019; <https://www.nature.com/articles/s41588-019-0480-1>

4. *Turkestanica* has previously been described as the closest relative to oleracea (Hammer K 2001). In the current analysis, the sample size of *Turkestanica* is quite small (n=3), and it appears that at least two of the accessions are collected from the same physical location or population in Georgia. Compared to the hundreds of oleracea accessions, it seems that the sample sizes of the other two species is too small to make conclusions based on a PCA. Going back to the sex chromosome question, it is possible that a divergent sex chromosome is driving the PCA patterns for *turkestanica*: are these 3 *turkestanica* individuals the same sex, or a mix of males and females (the supplemental data shows "NA" for sex)? If all are the same sex, it is worth testing if sex is driving these PCA patterns, and would benefit from the suggestion in comment #1 to genotypically infer the sex of all resequenced individuals with known sex markers.

Response: We are sorry that the reviewer was confused by the two wild species. The number of accessions of *S. turkestanica* used in this study is 7. Three is the number of *S. tetrandra* accessions. We recorded sex

phenotype of all seven *S. turkestanica* accessions, of which four are male and three are female (Supplementary Table 14), indicating sex is not driving these PCA patterns. To further confirm this, we performed the analysis using SNPs excluding those from the sex chromosome (chromosome 4). A nearly identical PCA pattern (Fig. R1 below) to that of using SNPs from all six chromosomes (Fig. S11) was observed, indicating that sex is not driving the PCA patterns observed in this study.

While we agree with the reviewer that larger sample size would provide more robust results, unfortunately, as mentioned in our manuscript, these were all accessions available in the U.S. National Plant Germplasm System. However, we believe that data from these accessions can confidently support the conclusions derived from PCA (Line 185-190).

Figure R1. Principal component analysis of spinach accessions using SNPs excluding those from chromosome 4.

5. The supplemental information is incredibly clear and easy to understand. Thank you for taking the time to present it so well.

Response: Thanks.

Minor comments:

L44: basal is not the right word here, as it implies a hierarchy to phylogeny. Terms like “basal” reflect misconceptions about evolution and phylogeny.

Response: Thanks for pointing this out. We removed “, the basal order of core eudicots”.

L74: 15X coverage is not “deep sequencing”.

Response: We removed “deep”.

L180: Should say n=7 I believe

Response: Done. Thanks.

L184: I would consider this a “germplasm collection” rather than population.

Response: Done. Thanks.

L189: Please clarify this sentence, e.g. Does this mean that 56.5% of the total SNPs called are shared between oleraceae and turkestanica?

Response: This means that these “56.5% of the total SNPs” are SNPs among all oleraceae and turkestanica accessions. We have made this clear in the revised manuscript (Line 171-172).

L206: Are all three species able to inter-cross? Is there ongoing gene flow?

Response: Yes, *Spinacia oleracea* can intercross with both *S. tetrandra* and *S. turkestanica* but the hybrid between *S. oleracea* and *S. tetrandra* has dramatically reduced fertility (Fujito et al., 2015; <https://doi.org/10.1534/g3.115.018671>). Gene flow analysis normally requires at least four different populations with large sample sizes. Due to the limitation of the available accessions (three populations and limited wild accessions), we are sorry that we were unable to perform a reliable analysis to identify gene flow between cultivated and wild spinaches.

L223: Define population fixation index here.

Response: Done. Thanks.

L277: Plant type/form is introduced in the manuscript without any description. What is type/form?

Response: Plant type refers to whether the plant is erect, semi-erect or spreading. This was defined in the Methods section (Line 497-500). We have revised this sentence to make it clear (Line 251).

L372: I would reserve the word “cutting-edge” for the latest technology, which I do not believe this assembly qualifies for. The manuscript is unclear on what PacBio platform was used, but it does not appear to be HiFi.

Response: PacBio CLR reads were used in the assembly. We have changed “the cutting-edge sequencing and assembling technologies” to “long-read and Hi-C sequencing technologies” (Line 337)

L380: Please clarify this sentence. What selective force is acting upon the sex chromosome during the evolution of Chenopodiaceae? Are other species in the family dioecious? Is there a shared origin of the sex chromosome with other species in the family?

Response: Spinach is the only dioecious species in the Chenopodiaceae mentioned in this study; therefore, sex chromosome in spinach arose relatively recently. We thank the reviewer for pointing this out and apologize for the potential confusion caused by this sentence and have deleted it in the revised manuscript.

L422: What PacBio sequencing platform was used? CLR? Which machine — RSII, Sequel-I, Sequel-II?

Response: This information has been added (Line 83 and 383-384).

Reviewer #2

The authors did excellent job for the research and for the manuscript! From the manuscript, authors create a new chromosome-scale reference genome assembly of spinach-inbred line ‘Monoe-Viroflay’ with a total size of 894.3 Mb and 98.3% anchored and ordered on the six chromosomes. The new spinach ‘Monoe-Viroflay’ genome assembly reference has higher genome coverage than those published in public, as I know

it is the highest coverage so far. Authors conducted genome-wide association studies of 20 agronomical traits and identified numerous significantly associated regions and candidate genes for the 20 traits, which will be useful in spinach molecular breeding. I recommend this manuscript to be accepted and published in Nature Communications.

Comments are

1. For the section “Planting and phenotyping” from lines 526 to 573, it will be nice to add some detail for phenotyping, such as how many replicates was used for the randomized complete block design (RCBD) design? A total of plants were used to do phenotyping for each trait or how many plants were used in each replicate.

Response: Thanks for the suggestion. The randomized complete block design was performed with three replicates, and 10 representative plants were randomly selected for the phenotyping of agronomic traits for each accession. We have made this clear in the revised manuscript (Line 493-495).

2. It will be nice to do some statistics analysis (such as ANOVA, distribution, correlation) for those quantitative phenotypic data such as plant height, plant width, leaf length, leaf width, leaf number, oxalate content, and downy mildew (DM) incidence and disease index (DI).

Response: Thanks for the suggestion. We have provided some statistics of the traits such as range, mean and coefficient of variation (Supplementary Table 15) and performed correlation analysis of these traits (Supplementary Table 16). We have added a short description of these results in the revised manuscript (Line 215-221).

3. For Fig. 3 of line 192-193 on page 9, the two-dimension PCA plot seems there is only one cluster (group) and almost all spinach genotypes (lines) merge at one place (3b) and the similar figure of the Supplementary Fig. 9 seems something wrong. For the 3c, it will be nice to add the cluster (sub-population) information from Fig 3c to the Table S10 for each of the 305 spinach accessions when K=2, 3, and 4 in order for readers to see each spinach accession belongs to which cluster (group) easily.

Response: Thanks for the suggestion. We have added the cluster/group information in Supplementary Table 10.

Both Fig. 3b and Supplementary Fig. 9 have no problem. Due to the very close relationship between *S. oleracea* and *S. turkestanica* and highly abundant *S. tetrandra*-specific SNPs in our dataset, all *S. oleracea* and *S. turkestanica* accessions could not be separated in the PCA plot shown in Supplementary Fig. 9. Fig. 3b (the PCA plot) actually shows another cluster comprising *S. tetrandra* accessions (bottom left of the figure). This PCA plot was generated using SNPs excluding those specific to *S. tetrandra*, and could separate *S. oleracea* and *S. turkestanica* accessions to a certain degree. This has already been explained in our manuscript (Line 185-190).

4. For Supplementary Fig. 11 “Linkage disequilibrium (LD) decay pattern of cultivated spinach (*Spinacia oleracea*)”, it will be nice to have six LD decay patterns by each chromosome, respectively!

Response: Thanks for the suggestion. We have added the LD decay patterns for the six chromosomes (Supplementary Fig. 14), all of which are largely similar to that of the entire genome.

Reviewer #3

Cai, Sun, Xu et al. present a comprehensive manuscript regarding spinach genome, evolution,

domestication and genetic basis of agronomic traits. They built a significantly improved spinach reference genome using state-of-the-art technologies which represents an important resource for applied and basic plant research. Using this resource, they notably performed macro-evolutionary analyses at the karyotype level. They resequenced 305 cultivated and wild spinach accessions and provided insight into spinach genetic diversity and population differentiation, performing GWAS of 20 agronomical traits and identifying domestication sweeps in the spinach genome. Globally, this study provides valuable resources for facilitating spinach breeding as well as insights into spinach evolution and domestication.

Overall, I find this work relevant and well-conducted. The results presented here will be of immediate interest to people working on different research areas (evolutionary genomics, plant breeding research, population genomics...).

I have several requests/suggestions for clarification and improvement of the results.

-It looks clear that the spinach reference genome generated here is of high quality and is improved compared to the previous ones (Sp75, Spov3 and SOL_r1.1). However, the alignment of newly generated pseudomolecules against pseudomolecules previously generated would be appreciated. Indeed, the scientific community working with those previous versions would strongly benefit from that. A dot-plot representation like in Supplementary Fig.3 would give a relevant overview.

Response: Thanks for the suggestion. The dot plots between the Monoe-Viroflay genome and the other three spinach genomes (Sp75, Spov3 and SOL_r1.1) have been added in the revised manuscript (Supplementary Fig. 3), which indicate that the Monoe-Viroflay genome and the other three spinach genomes are largely collinear. However, due to the fragmented and less complete nature of the other three genomes, especially the Sp75 and Spov3 genomes that were assembled using Illumina short reads and reads from early PacBio technology (RSII), respectively, it's not surprising to see some small inconsistent alignments (mainly in highly repetitive region). This further highlights the value of the high-quality spinach genome assembled in this study.

-As a complement of Fig.1 b (in Supplementary Figures for example), it would be interesting to have plots showing the real numbers regarding gene, TE, SNP and SV densities (scatter or density plot for example). Indeed, heatmaps give a nice overview but are not precise for more advanced observations. Also, the chromosome painting (based on the ancestral karyotype) could be represented under the x-axis of those plots. This would provide direct visualization of the potential correlation between chromosomal rearrangements and gene, TE, SNP, and SV densities.

Response: Thanks for the suggestion. The figure has been added in the revised manuscript (Supplementary Fig. 2).

-Along the same line of the previous comment, it would be interesting to check whether fission and fusion points show particular gene or TE patterns. For example, spinach chromosome 4 (protochromosome 1) which has been shown to be very stable during Chenopodiaceae evolution seems to exhibit high gene density in the two subtelomeric regions, which does not seem to be the case for the other chromosomes. One could wonder how gene and TE densities are conserved across species depending on rearrangements. There is maybe something interesting to dig into in this context.

Response: We checked gene and TE patterns around fission and fusion points and couldn't find anything interesting. In addition, in the revised manuscript we have deleted the sentence "We found that most of Chenopodiaceae protochromosomes have undergone fission and fusion during the path to spinach, but the protochromosome 1, corresponding to spinach chromosome 4, has been largely maintained (Supplementary

Fig. 2). Both our GWAS and previous study²⁹ have revealed chromosome 4 as the sex chromosome, suggesting the selective force acting on this chromosome during the evolution of Chenopodiaceae species” (Please check our response to the second to the last comment of Reviewer 1), thus making this point moot.

-In Fig.2, it would be relevant to represent Quinoa and Amaranth WGDs as the gamma triplication is represented. Indeed, those additional and specific WGDs directly explain the chromosome number expansion for those species.

Response: Fig. 2 has been updated accordingly. Thanks.

-It is difficult and not accurate to assess it by eyes on the figures, but I am wondering whether the most likely ancestral karyotype is composed of 9 or 10 chromosomes. I am wondering whether the beginning of the Sugar beet chromosome 1 corresponding to the end of the garden orache chromosome 9 should be an independent ancestral chromosome or combined with the other purple regions. Could you please check it out and provide pieces of evidence?

Response: The most likely ancestral karyotype is composed of 9 chromosomes, as shown in the top left of Fig. 2. The beginning of the Sugar beet chromosome 1 corresponding to the end of the garden orache chromosome 9 should be derived from the same ancestral chromosome combined with the other purple regions, supported by the synteny between sugar beet and ancestral Chenopodiaceae genomes as shown in Supplementary Fig. 5.

-line 379: You probably mean Supplementary Fig. 3, isn't it?

Response: This sentence has been removed in the revised manuscript as indicated above.

Reviewers' Comments:

Reviewer #1:

Remarks to the Author:

The authors have sufficiently and accurately responded to all of my comments; thank you for taking the time to thoroughly respond.

Reviewer #2:

Remarks to the Author:

Dear Authors,

Thank you for your thorough response to the comments from reviewers. I think that your revised manuscript is much improved.

I recommend this version of your manuscript to be accepted and published in Nature Communications.

Best regards!

Ainong

Reviewer #3:

Remarks to the Author:

The authors answered all my comments and suggestions. Congratulations!

Reviewer #1

The authors have sufficiently and accurately responded to all of my comments; thank you for taking the time to thoroughly respond.

Response: Thanks.

Reviewer #2

Thank you for your thorough response to the comments from reviewers. I think that your revised manuscript is much improved. I recommend this version of your manuscript to be accepted and published in Nature Communications.

Response: Thanks.

Reviewer #3

The authors answered all my comments and suggestions. Congratulations!

Response: Thanks.